# Application of Radar Solutions for the Purpose of Bird Tracking Systems Based on Video Observation

**DOI:** 10.3390/s22103660

**Published:** 2022-05-11

**Authors:** Ksawery Krenc, Dawid Gradolewski, Damian Dziak, Adam Kawalec

**Affiliations:** 1Bioseco S.A., Budowlanych 68, 80-298 Gdansk, Poland; dawid.gradolewski@bioseco.com (D.G.); damian.dziak@bioseco.com (D.D.); 2Faculty of Mechatronics, Armament and Aerospace, Military University of Technology, Sylwestra Kaliskiego 2, 00-908 Warsaw, Poland; adam.kawalec@wat.edu.pl

**Keywords:** bird tracking, tracking algorithms, camera detection, visual sensor network, wildlife hazard management, stereo-vision

## Abstract

Wildlife Hazard Management is nowadays a very serious problem, mostly at airports and wind farms. If ignored, it may lead to repercussions in human safety, ecology, and economics. One of the approaches that is widely implemented in small and medium-size airports, as well as on wind turbines is based on a stereo-vision. However, to provide long-term observations allowing the determination of the hot spots of birds’ activity and forecast future events, a robust tracking algorithm is required. The aim of this paper is to review tracking algorithms widely used in Radar Science and assess the possibilities of application of these algorithms for the purpose of tracking birds with a stereo-vision system. We performed a survey-of-related works and simulations determined five state-of-the art algorithms: Kalman Filter, Nearest-Neighbour, Joint-Probabilistic Data Association, and Interacting Multiple Model with the potential for implementation in a stereo-vision system. These algorithms have been implemented and simulated in the proposed case study

## 1. Introduction

Recent research shows that in 2020 alone more than 10,000 bird collisions with airplanes were reported, putting the lives of more than 2 million passengers at risk [1]. Only in the USA, the average annual repair cost exceeded $200 million [2]. The bird-strikes are generally associated with three aspects: human safety, ecology, and economics [3]. All of which are imperative to consider when designing modern Wildlife Hazard Management (WHM) systems for airport runways, more often requested by the International Civil Aviation Organization (ICAO) and European Union Aviation Safety Agency (EASA) [4,5]. Radar- and camera-based solutions are widely implemented as a base for such systems [3]. While the radar sensors allow the monitoring of vast runway areas in all weather and light conditions, the vision-based systems are applied for less challenging applications as the cheapest alternative, allowing object classification during the day-light operations [3].

State-of-the-art camera-based bird detection systems [6,7,8,9] are complex sensor networks allowing the detection of the bird and identification of the event. However, for in-depth analysis allowing to determine the potential threat early in advance, the localization and tracking of the birds’ activity is essential. This is necessary to provide long-term observations allowing the determination of the hot-spots of birds’ activity and forecasting future events at the airport runways. Therefore, tracking birds is nowadays a very important and urgent problem.

It was shown that large-base stereo-vision is a method allowing the determination of the object’s state vector in the 3D space with relatively small positioning error enabling reconstruction of birds’ trajectories [3]. The trajectory reconstruction could be performed by a tracking algorithm. In Radar Science, this problem is well known [10,11], where tracking filters powered by the raw radar data are used to initialize the plots. A publication of a tracking estimator by Rudolf Kalman in the 1960s[X], known today as the Kalman Filter, became a catalyst for the formation of the new tracking filters. Over time, these solutions have strengthened their position in Radar Science, finding applications in many military and civil systems.

In this paper, we applied the well-known radar-based tracking algorithms into the stereo-vision based airports monitoring system. The state-of-the art tracking algorithms: Kalman Filter, Nearest-Neighbour, Joint-Probabilistic Data Association, and Interacting Multiple Model, have been selected and implemented in a Matlab software. The systems have been heuristically optimized with the use of the AFMS system [3,6] and bird-like drones equipped with a GPS sensor. It was shown that besides some differences at the senor level, such as the detection range, measuring uncertainty, sampling, and false detection rates; the data processing in both cases is the same and the radar tracking algorithms could be easily adapted for WHM.

## 2. Background and Related Works

Since Rudolf Kalman’s publications on linear filtering and prediction in 1960 [12,13,14], a fast development of estimation techniques has been started. Algorithms such as IMM, Interacting Multiple Model, and PDA(F), Probabilistic Data Association (Filter), have been designed in order to enhance the estimation process. They were dealing with problems such as lack of knowledge on process noise or false detections, which normally emerged whenever the Kalman Filter was to be applied in the real world. In most cases, the above-mentioned techniques and algorithms have been used for tracking relatively big objects such as ships and airplanes, based mainly on radar measurements. It is estimated that in the beginning of the 1990s the majority of the surveillance systems in the US were equipped with tools such as the Kalman Filter in order to support the operator in revealing the true trajectories of the observed objects.

The further development of sensors and application of multiple sources has fostered the modernization of the known techniques and algorithms, e.g., JPDA, Joint Probabilistic Data Association [15,16,17], as well as inventions of new ones, such as MHT, Multiple Hypothesis Tracker [18,19,20,21,22] or PHD, Probability Hypothesis Density [23,24,25,26]. With new sensors such as stereo video cameras, tracking smaller objects such as drones and birds has become possible.

Nowadays, the estimation tools based on Kalman Filtering for tracking birds are commonly used in surveillance systems, as well as in wind turbine protection systems [27,28]. There are PhD and MSc theses on the application of these techniques for tracking birds and other animals [29,30]. On the other hand, the advanced tracking algorithms such as JPDA, MHT or PHD are still evolving in order to meet new requirements of systems for resolving many different tracking and information fusion problems.

The most common bird tracking methods consider: Radio tracking, Satellite tracking, Geolocator loggers, RFID or GSM tags [31].

## 3. Problem Statement and Objectives

As the survey-of-related works shows, there is a lack of suitable tracking algorithms for large-base stereo-vision systems designed to surveil the vast monitoring area (in a radius of 500 m). Nonlinear uncertainty measurement errors, non-deterministic time sampling rates, and false-positive detections [3,32] are the potential problems with the implementation of radar-based tracking algorithms.

The main objective of this paper is to present a survey of possible solutions and to determine which radar tracking algorithms can be used for the precise and robust bird tracking of large-base stereo-vision systems. The heuristic optimization of the algorithms, as well as system validation are within the scope of the research.

Based on the performed survey and initial simulation four radar tracking algorithms: Kalman Filter, Nearest-Neighbour, Joint-Probabilistic Data Association, and Interacting Multiple Model have been selected and implemented in the Matlab software. The measurements performed in the real environment with the use of the AFMS system [3,6] and bird-like drone equipped with a GPS sensor have been used for the heuristic optimization of developed solutions.

## 4. System Description

Due to the specificity of the research work, the system description section has been divided into the following subsections:data collection system;tracking algorithm testbed.

The first section discusses the structure and core parameters of the system used for data acquisition, while the second describes an application created for testing the tracking algorithms.

### Data Acquisition System

The data acquisition system used in this work is based on the stereo-vision concept described in [3,6].

As presented in Figure 1, the system is based on the distributed computing concept embedded in the IoT paradigm. The proposed solution provides real-time processing on monitoring modules and decision making on the system control unit. Moreover, the proposed approach, apart from data processing, allows easy data handling and presentation to the final user [3,6,32].

The processing architecture of the proposed solution is presented in Figure 2. Each monitoring module is equipped with a dedicated CPU, processing video stream from cameras coupled in stereo-vision set. Each frame is processed to detect movements and then only detected objects are subjected to the identification bird process as presented in [32]. Each detection classified as a bird is processed further with information about the object’s center coordinates, width, height and contour size. Next, the estimation of the object’s 3D localization is performed [3,6,32].

Localization data, along with object characteristics and miniatures, are further sent via Ethernet to the control unit. They are analyzed statistically in the data filtering block to filter out non-bird objects remaining after identification. Only the above-prepared data are sent to the BPS Tracking module which is a key component described in this work. The data are aggregated into the individual tracks for each detected object. Further, track data together with object width, height and contour received from the monitoring module are fed to the size classification algorithm where object size is estimated similarly to that in [3,32]. Depending on the estimated size and track of the bird, the decision about deterrent is made and all data are stored in a dedicated database. Then, they are available to the user via web-based HMI [3,6,32].

## 5. Examined Tracking Algorithms

The examined algorithms have been divided into several categories, with respect to the following features:object motion model;object observation model;multiple model interaction.

### 5.1. Object Motion Model

The ideal tracking algorithm should be able to track manoeuvring objects properly regardless of their kinematics. No matter if the object is stationary (and slowly going along the line) or highly manoeuvring (and doing turns) its state vector is always retrieved with estimation errors on a satisfactory level. Unfortunately, this cannot be technically achieved without any prior knowledge of the objects’ kinematics. Therefore, one of the most important parts is object motion modeling.

#### 5.1.1. Simple No-Model Tracker

Fortunately, in many tracking problems the subject of interest does not cover the whole spectrum of kinematics, but its narrow fraction. In other words, we are often interested in tracking only specific objects of certain manoeuvrability. This may be regarded as a sort of suppression of the role of the object motion model, and perform a kind of justification for tracking filters which include only a constrained form of the object motion model, like α-β filters, or do not include that model at all.

For the purpose of the considerations presented in this work, we chose to include in our comparison a simple *ad hoc* tracker with no object motion model installed. In this filter, the resulting state estimate is a mixture of the current measurement and the state estimate taken from the previous step as the Formula (Equation 1) shows.
(1)$$x\left(k+1\right)=\left(1-r_{m}\right)\cdot x\left(k\right)\cdot M_{3\times 6}+r_{m}\cdot z\left(k+1\right)$$
where:

x(k): object state vector in the spherical coordinate system;

z(k): measurement in the spherical coordinate system;

$M_{3\times 6}$: mask matrix;

$r_{m}$: measurement to previous state ratio.

In numerical experiments, we accepted the object state vector to be defined in the spherical coordinate system. This is mainly due to compatibility with other examined tracking filters described in the following subsections. Taking into account two dynamic coordinates (position and velocity), the parameters of the filter may be defined as follows:(2)$$x\left(k\right)=\left[r,\dot{r},\phi ,\dot{\phi },\theta ,\dot{\theta }\right]$$
(3)z(k)=[r,ϕ,θ]
(4)$$M_{3\times 6}=\left[\begin{array}{cccccc} 1 & 0 & 0 & 0 & 0 & 0\\ 0 & 0 & 1 & 0 & 0 & 0\\ 0 & 0 & 0 & 0 & 1 & 0 \end{array}\right]$$
(5)$$r_{m}=0.15$$

#### 5.1.2. Spherical Kalman Filter

Even though for most of existing radars the spherical coordinate system performs a default system, in the radar literature on tracking estimators, the Kalman Filter is very often expressed in Cartesian coordinates [11].

However, if the sensor measurements are expressed in the spherical system, the more natural it is to provide an estimation in the spherical system. Apart from the simplicity of such a solution, we may significantly suppress the measurement error propagation which may differ for the subsequent spacial coordinates. For example, for the used camera, the range error is significantly higher than the azimuth and the elevation errors. Performing an estimation in the spherical coordinates allows to supply the tracking filter with angle data of good quality. Otherwise, the range errors would affect all three dimensions due to the transformation from the spherical to Cartesian coordinate system.

Therefore, for the purpose of the presented considerations, we chose to include the Kalman Filter defined in the spherical coordinate system. Similarly, as for the no-model tracker, we assumed the state vector to be defined according to (Equation 2).

Matrices of the state model used by the Kalman Filter in all of the numerical experiments have been defined according to [10,11], as follows:

State-transition matrix *A*(*k*):(6)$$A\left(k\right)=\left[\begin{array}{cccccc} 1 & T & 0 & 0 & 0 & 0\\ 0 & 1 & 0 & 0 & 0 & 0\\ 0 & 0 & 1 & T & 0 & 0\\ 0 & 0 & 0 & 1 & 0 & 0\\ 0 & 0 & 0 & 0 & 1 & T\\ 0 & 0 & 0 & 0 & 0 & 1 \end{array}\right]$$

Stochastic input matrix *B*(*k*):(7)$$B\left(k\right)=\left[\begin{array}{ccc} T^{2}/2 & 0 & 0\\ T & 0 & 0\\ 0 & T^{2}/2 & 0\\ 0 & T & 0\\ 0 & 0 & T^{2}/2\\ 0 & 0 & T \end{array}\right]$$
and the process noise covariance *Q*(*k*):(8)$$Q\left(k\right)=\left[\begin{array}{cccccc} \sigma_{r}^{2}T^{4}/4 & \sigma_{r}^{2}T^{3}/2 & 0 & 0 & 0 & 0\\ \sigma_{r}^{2}T^{3}/2 & \sigma_{r}^{2}T^{2} & 0 & 0 & 0 & 0\\ 0 & 0 & \sigma_{\varphi }^{2}T^{4}/4 & \sigma_{\varphi }^{2}T^{3}/2 & 0 & 0\\ 0 & 0 & \sigma_{\varphi }^{2}T^{3}/2 & \sigma_{\varphi }^{2}T^{2} & 0 & 0\\ 0 & 0 & 0 & 0 & \sigma_{\theta }^{2}T^{4}/4 & \sigma_{\theta }^{2}T^{3}/2\\ 0 & 0 & 0 & 0 & \sigma_{\theta }^{2}T^{3}/2 & \sigma_{\theta }^{2}T^{2} \end{array}\right]$$
where: $\sigma_{r}$, $\sigma_{\varphi }$, and $\sigma_{\theta }$ denote parameters describing process noise standard deviations for the spherical coordinates: range, azimuth, and elevation, respectively.

For simplicity, in the numerical experiments it was assumed:(9)$$\sigma_{r}=\sigma_{\varphi }=\sigma_{\theta }=\sigma =0.01$$

### 5.2. Object Observation Model

The object observation model is basically the component of the filter related to measurements and measurement accuracy. Since the number of the state vector coordinates differs from the number or observation vector coordinates (see (Equation 2) and (Equation 3)), a respective observation matrix has to be defined. In the numerical experiments, we assumed it to be defined in the same manner as for the no-model tracker $C\left(k\right)=M_{3\times 6}$:

Whether the observations originate from sources of relatively good or bad quality, it should be reflected by appropriate values of the observation model parameters. One of these parameters is the observation noise standard deviation. Keeping in mind that due to the diversity of errors for each of the spacial coordinates, we expressed the object kinematics in the spherical coordinate system may decompose this standard deviation into three parameters: Δr, Δϕ, and Δθ, referring to the accuracy of: range, azimuth, and elevation, respectively.

For the numerical experiments, a vector of the observation noise standard deviation has been defined empirically as follows:(10)$$\left[\begin{array}{ccc} \Delta r & \Delta \varphi & \Delta \theta \end{array}\right]=\left[\begin{array}{ccc} 3.5 & 0.5 & 0.5 \end{array}\right]$$

However, the accuracy of measurement is not the only parameter of the observation model that is taken into account. Due to the fact that some of the detection may not originate from the tracked objects, there is a need for the observation model to enable measurement clustering. Additional processing of the measurements associated in each of the clusters allows (depending on the particular algorithm) either to choose or to elaborate (by, e.g., probabilistic association) the most appropriate detection. This performs the basis for the functioning of such algorithms such as Nearest-Neighbour or Joint Probabilistic Data Association, described below.

#### 5.2.1. Nearest-Neighbour Tracker

Nearest-Neighbour (NN) is simpler than that mentioned above, and it may be regarded as an algorithm that enhances the performance of the Kalman Filter. In this algorithm, detections (gathered in clusters) undergo evaluation based on the distance metric calculated in reference to the state prediction obtained from the motion model. In each step of the Kalman filtering, the detection which is the closest is accepted as the observation Z(k) and being processed further according to the Kalman estimation formula.

Due to the fact that the observation vector Z(k) is selected from all the detection within a single cluster, this solution is often classified as exclusive.

#### 5.2.2. Joint Probabilistic Data Association

Joint Probabilistic Data Association (Filter), JPDA(F), is an extension of Probabilistic Data Association, PDA, algorithm allowing to track more than one object within a given space. On the contrary to NN, this algorithm is not exclusive. That means all the detections within a single cluster are effectively utilized in elaboration of the resulting observation Z(k).

Figure 3 presents the block diagram of the PDA algorithm including the particular steps of Kalman filtering for the reader to have a notion of how these two algorithms cooperate with each other.

Based on that, the reader may notice that each of the detections participates in elaboration of the resulting observation Z(k), and the contributions of the particular detections are determined upon the calculated association probabilities.

The JPDA algorithm is very similar in that sense. The difference resides in that all the measurements are being associated with each of the tracked objects. Thus, for example, if some of the detections do not originate from a particular object, the algorithm assigns relatively small association probabilities to these detections. Finally, due to the non-exclusive nature of the filter, these detections contribute in the resulting observation Z(k), although their contribution is insignificant.

The fundamental assumption which holds whenever applying any of the PDA algorithms is that the number of the tracked objects is known *a priori* to the algorithm. In the case of PDA, it is simply one. In case of JPDA, it is typically more than one, however, it should be a reasonably small number in order to prevent from potential numerical explosions.

### 5.3. Multiple Models Interaction

As it was mentioned in the Section 5.1, knowledge of kinematics of the manoeuvring object is often crucial to effectively tracking these objects. Then, the following question may be raised: “What happens if we do not have such knowledge, or if we need to track multiple objects of different kinematics at once?”

The Interacting Multiple Model (IMM) algorithm seems to be a good solution for the above problem. Similarly to PDA, the IMM algorithm may act as a solution for enhancing Kalman Filter performance. However, on the contrary to PDA, IMM works within the scope of object motion modeling, not object observation modeling.

The basic idea that stands behind that algorithm is: instead of sticking to a particular model, a couple of models with an instruction on how to operate these models can be used. Particularly, in IMM, the models’ operation is a mixing formula. This means that in every KF prediction step each of the defined models is always active.

Figure 4 shows the simplified block diagram of the IMM algorithm. Based on that diagram, there are two stages of mixing local estimators $x_{1}\left(k\right),\ldots x_{n}\left(k\right)$: the first is before the Kalman filtering, and the second one is after the Kalman filtering and calculation of conditional probabilities.

If the object kinematics is unknown, i.e., its motion model cannot be defined precisely, the IMM algorithm with at least two models may be applied. In such a case, the first model refers to the hypothesis of a highly manoeuvring object, while the second one refers to the hypothesis of a relatively stationary object. During tracking, each of the local estimators produces its own estimates and covariances, and the IMM algorithm fuses these two pairs into a global (resulting) state estimate and covariance. The particular contribution of each of the local estimators vary in time depending on the actual kinematics of the manoeuvring object.

It is important to notice that, according to Figure 4, IMM does not constrain the number of interacted models to two. We can apply as many local estimators as we require. That feature is very useful, especially if we desire to apply some very particular motion model for specific objects.

The simplest use case of the IMM algorithm requires at least a standard Kalman Filter. This is due to the fact that in order to interact multiple models we simply need to have these models defined. Furthermore, the calculation of innovations and Kalman gains is necessary for the evaluation of these local models. Thus, in the literature, the combination of IMM and KF usually shortens to IMM. However, for the sake of the considerations presented in this paper, we clearly state that combination in order to distinguish it from the spherical Kalman Filter, or from any other combination of IMM algorithm presented in the following subsections.

It is important to notice that in our numerical experiments we have been using IMM as a solution for unknown standard deviation of the process noise. Two structurally identical motion models have been defined. The only difference resided in predefined values of the models’ process noise standard deviations. The particular values of the standard deviations have been defined empirically after numerical experiments conducted prior to those described herein.
(11)$$\left\{ \begin{array}{c} \sigma_{M1}=0.001\\ \sigma_{M2}=0.1 \end{array}\right.$$

As it was pointed out in Section 5.2, Nearest-Neighbour is a simple algorithm enabling the enhancement of the Kalman Filter. Due to the fact that it operates in the scope of the observation model, there are no contradictions to apply it together with IMM, which operates in the area of the object motion model.

Similarly, as in IMM, KF in the numerical experiments IMM performed as a tool to combine two identical (in terms of structure) motion models with diverse models’ standard deviations. Similarly to NN, the JPDA algorithm can be applied together with IMM. For the purpose of the numerical experiments, it was used with the same structure and parameters as IMM NN.

An interesting hybrid of the two algorithms mentioned above is a combination of IMM, NN, and JPDA. In that combination, IMM was used for interacting NN and JPDA local filters. The summary of considered tracking algorithms is presented in Table 1.

## 6. Filter Comparison and Testing Methodology

For the purpose of the considerations presented in this paper, a number of numerical experiments have been performed. As it was mentioned in the introduction, all these filters are applicable in radar solutions on a regular basis, which means their features are well-known to the designers of surveillance systems or command and control systems. Therefore, our intention was to examine how these algorithms fit into our needs, and particularly, to what degree we may use them for effective tracking of the birds detected by the stereo-vision system, rather than extensive and comprehensive analysis of each of the algorithms.

For the above-mentioned reason, we have organized the experiments into the following three stages:Stage 1: single bird scenario;Stage 2: multiple bird scenario;Stage 3: drone flight scenario.

In the first two stages, we have compared the performance of the tracking filters using collections of archived bird flight scenarios, which had been recorded prior to the numerical experiments. As the observations refer to wild animals in nature, for obvious reasons, we do not possess the ground truth data to refer performing comparisons. Thus, tracking filters have been assessed subjectively based on the quality of the reconstructed bird trajectories, taking into account such aspects as completeness, continuity, and smoothness of the bird tracks.

In the third stage, we have compared the performance of the tracking filters using collections of recorded drone flight scenarios. As the drone had been equipped in GPS transmitter the aim of the experiment was to compare the filters’ estimates in reference to ground truth - the GPS location data.

In this section, we present the results of the performed numerical experiments run for each of the above-described filters. The examined trackers have been compared using the same scenario.

### 6.1. Stage 1: Single Bird Flight Scenario

In the first stage, we have selected the most representative set of detections referring to a single bird flight. As one may notice, relatively high observation noise variation in comparison to the next presented sets of detections. The scenario is not extremely difficult for the tracking filters in terms of the bird motion kinematics, but as the observation noise covariance is high the measurements are pretty much scattered. Thus, we may expect the results will reveal dominance of NN and JPDA filters over the rest of the filters not equipped with these algorithms.

For better clarity of the presented comparison, the results have been associated in two categories referring to: known process noise covariance and unknown process noise covariance.

#### 6.1.1. Filters with Known Process Noise Covariances

In the first experiment, no-model tracker has been run. Figure 5 (black x symbols) presents the performance of that filter.

According to that run, it is not difficult to see that the depicted detections indicate quite a significant variance, which is typical for observations using the stereo-vision. Nevertheless, even the most basic filter (of the herein considered) enabled to reconstruct the bird’s trajectory as a continuous track. That track is, however, significantly twisted which is obviously a consequence of not having an existing object motion model, but despite that, it provides some reasonable and useful data for further processing in the system.

In the second experiment, the spherical Kalman Filter has been run. Figure 5 (red x symbols) presents performance of that filter.

Compared to the no-model solution, the spherical Kalman Filter enables a noticeable smoothing of the reconstructed trajectory. This is clearly visible, particularly, in the second half of the track. However, at the beginning of the track we may observe a significant susceptibility to scattering of the measurement data. That, on the other hand, is also characteristic of solutions based on the Kalman filter, and results from the initialization phase, where only after a few steps can the full state vector be set, and the motion model is ready for operation.

In the next experiment, the Kalman Filter with the Nearest-Neighbour enhancement has been used. As in the NN algorithm only the closest (to the predicted state) detection is taken for further processing, before starting the experiment we expected the resulting trajectory would be noticeably smoothed, especially in the beginning phase.

Figure 5 (green x symbols) shows the performance of the NN algorithm. It is not difficult to notice that the whole trajectory is very significantly smoothed compared to either the no-model tracker or the spherical Kalman Filter. This is mainly the effect of two factors. The first one is clustering in general, i.e., detections before get “digested” by the Kalman Filter undergo processing. The second is a particular way of that processing, which by application of the NN algorithm performs the most (of herein considered) softening one.

The last examined filter in the group of estimators with known process noise covariance was Joint Probabilistic Data Association. Figure 5 (blue x symbols) presents the performance of that filter.

Compared to the previously examined filters, this one, similarly to NN, is distinguished by softening the stationary fragment of bird’s trajectory. However, unlike NN, it emphasizes the bird’s manoeuvre in the first half of the track. Thus, based on Figure 5, we can conclude that this filter seems to be an excellent compromise between smoothing out strongly distorted measurement data and reflecting the kinematics of the flying bird.

#### 6.1.2. Filters with Unknown Process Noise Covariances

In the second phase of the numerical experiments, tracking filters with no prior knowledge about the process noise covariance have been tested. A combination of Interacting Multiple Model and the spherical Kalman Filter algorithms was the first choice. Figure 6 (red x symbols) presents the performance of that filter.

As the IMM algorithm was used as a tool to solve the problem of not knowing the covariance of the process noise, in our opinion the most reasonable would be to compare the results of these filters with the results of their analogues with the known process noise covariance. Thus, comparing the performance illustrated in Figure 6 (red x symbols) to the performance shown in Figure 5 (also red symbols), we may comment that they are practically identical. That means the IMM algorithm with the given settings works inline with our expectations.

Therefore, we may presume that in practical situations even if we do not know the exact kinematics of the manoeuvring object with IMM we would probably be able to reconstruct the bird’s trajectory for similar scenario.

In the next two experiments, the IMM versions of NN and JPDA algorithms have been tested. Figure 6 (green x symbols and blue x symbols) shows that they are almost identical with their “non-IMM” analogues. Practically, the only noticeable difference is that in the case of NN, for the IMM version the reconstructed trajectory is slightly shifted to the right, so all of the very last detections (around x = −135 m, y = −30 m) can be seen on the left side of the reconstructed trajectory, while in the case of NN with known process noise covariance the trajectory passes between these detections (see Figure 5).

In the last performed experiment, the combination of IMM with NN and JPDA algorithms have been tested. Since the research conducted so far showed that the NN algorithm has a tendency for over-smoothing trajectories, we expected that a combination of NN and JPDA may be very useful. Namely, JPDA would dominate in case of a manoeuvre, while NN would take over in the case of a stationary fragment of the bird’s trajectory.

Figure 6 (purple x symbols) shows that to some extent we were right. However, surprisingly (around x = −117 m, y = −88 m) one may observe a jump in the trajectory caused by a sudden “change” of motion models. Fortunately, that jump did not cause a break in the track.

### 6.2. Stage 2: Multiple Bird Flight Scenario

In the second simulation stage, we have selected the set of detections referring to multiple bird flights. The difficulty of that scenario resides in the kinematics of the bird motion. The actual task was to reconstruct two trajectories. The first one was long, and refers to a bird flying towards a wind turbine, and then making a manoeuvre to fly away. The second one was short, and refers to some bird flying far from the wind turbine.

Similarly to stage 1, for better clarity of the presented comparison, the results have been associated with two categories referring to: known process noise covariance and unknown process noise covariance.

#### 6.2.1. Filters with Known Process Noise Covariances

In the first experiment, the no-model tracker has been run. Figure 7 presents the performance of that filter. According to that figure, it is not difficult to notice that the tracker performance was very satisfactory. Both tracks have been reconstructed completely, continuously, and smoothly. As it was explained in Section 5.1.1, no-model filter performs a mixture of the current measurement and the previous measurement with the fixed ratio. Thus, its performance depends only on the quality of the gathered detections.

In the second experiment, the spherical Kalman Filter has been applied. Figure 8 presents the performance of that filter. Similarly, as for the no-model solution, the spherical Kalman filter enables the reconstruction of smooth, and mostly complete trajectories. The shorter trajectory has been reconstructed slightly better than that constructed by the no-model tracker. However, the longer trajectory introduces noticeable deviations of the state estimate in reference to measurements caused by the motion model. That is perfectly visible in the moments of the manoeuvre, especially in the location (X = 50, Y = −150), where the continuity of the track has been broken.

In the next experiment, the Kalman Filter with the Nearest-Neighbour enhancement has been used. Figure 9 shows the performance of the NN algorithm.

It is not difficult to notice that the observed performance of that filter is unsatisfactory for the WHM system. For the longer trajectory, in the first stage of tracking the filter seems to ignore measurements (except the initial one) and provides a track directly according to the motion model. After the manoeuvre around (X = 300, Y = −450) the filter begins to estimate stably, but in the location (X = 50, Y = −4150) estimation degrades as the filter tries to initiate multiple tracks. The shorter trajectory is not reconstructed at all. The respective track consists of only one point—the initial measurement.

The last examined filter in the group of estimators with known process noise covariance was Joint Probabilistic Data Association. Figure 10 presents the performance of that filter. Based on that figure, we may observe that the filter is noticeably better than the NN Kalman Filter. For the longer trajectory, the deviations visible in the first stage of tracking are substantially smaller. Although, after the manoeuvre in the location (X = 50, Y = −150), the filter breaks the track and the track is reinitialized even faster than in the case of the spherical Kalman Filter. However, the shorter trajectory is reconstructed partially. Even though the filter provides better results than the NN Kalman Filter its performance is still hard to classify as satisfactory.

#### 6.2.2. Filters with Unknown Process Noise Covariances

In the second phase of the numerical experiments, tracking filters with no prior knowledge about the process noise covariance have been tested. A combination of Interacting Multiple Model and the spherical Kalman Filter algorithms was the first choice. Figure 11 presents the performance of that filter.

The Kalman Filter in comparison to the IMM looks slightly better. Both trajectories have been reconstructed smoothly without significant deviations in the moments of manoeuvre. Although in the location (X = 50 m, Y = −150 m) the filter breaks the track, however, it is instantly reinitialized. What is more, if comparing with the performance of all the previously examined estimators, the IMM Kalman Filter shows better model-based filtering.

Figure 12 presents the results of the IMM and NN comparison. As one may see, the filter provides substantially worse results, mostly in the beginning phase, and after the manoeuvre in the location (X = 50 m, Y = −150 m), where the track is significantly scattered. Additionally, the shorter trajectory is not reconstructed. Nevertheless, we may observe a significant improvement in smoothness and continuity of the longer part of the trajectory.

In the last experiment of the second stage, a combination of IMM and JPDA has been examined, what summarizes the Figure 13. The stationary fragments of the longer trajectory have been reconstructed correctly with no substantial deviations. As for all of the model-based algorithms, the filter broke down the track in the location (X = 50 m, Y = −150 m), however, the tracks reinitialization was shorter in time. The main drawback of this filter was the reconstruction of the shorter trajectory.

From the presented results a general conclusion may be drawn that: “The simplest filter gives the better performance”. This is in opposition to the primary assumption that no-model filters would perform better for low values of the observation noise variances. Furthermore, any fixed model interference caused an increase in the estimation errors. In the second stage of the experiment, the application of NN algorithms, as well as JPDA did not bring expected improvement. These algorithms have been designed to work in a noisy environment, however, when applying for the birds tracking using long-base stereo-vision obtained results are incoherent.

### 6.3. Stage 3: Drone Flight Scenario

In the last stage, we have used a bird-like drone equipped with the GPS sensor designed to perform a circular flight around the system, see [3]. The aim of the experiment was to obtain ground-truth data from the GPS sensor. It was necessary to transfer the data to a common coordinate system by adding a constant offset, see Figure 14.

Because the NN algorithm and JPDA did not provide suitable results, further simulations have been performed on the spherical Kalman and the IMM filters. Figure 15 presents the comparison of the estimation errors calculated for KF (red color) and IMM KF (blue color). The green color represents the measurement errors. As one can see, except for the moment around sample no. 90 where IMM algorithm provided a slightly smaller estimation error, the performance of all tested tools is very similar.

## 7. Discussion

An attempt to adopt a radar tracking algorithm for the long-base stereo-vision-based bird monitoring system succeeded. We performed a survey-of-related works and initial simulation determining five state-of-the-art algorithms: Kalman Filter, Nearest-Neighbour, Joint-Probabilistic Data Association, and Interacting Multiple Model with the potential for implementation. These algorithms have been implemented and simulated in a Matlab software.

The data from the stereo-vision-based Airport Fauna Monitoring System (AFMS) of Bioseco has been used for the simulation. Three case-study scenarios: single bird flight, multiple bird flight and simulation of the flight by the bird-like drone equipped with GPS sensor have been designed and implemented.

Table 2 summarizes the conclusion drawn from the simulations. During the first case scenario, all the filters handle the object tracking. Overall, the second and third case studies show that the spherical Kalman and the IMM filters outperform other solutions in terms of track continuity and trace smoothing. The NN and JPDA algorithms fail in terms of showing great discontinuity in determined tracks.

Although, the presented study shows the possibility for the implementation of the radar tracking algorithm in the Wildlife Hazard Management System based on stereo-vision, still more simulations are required to fully adopt the IMM and JPDA algorithms. The trajectory reconstructed by this filter in the stationary parts of the track is very smooth, while in its more dynamic parts the system breaks the tracks.

A quantitative validation is still required before implementing the tracking algorithm in a real environment at the airports’ runway. The authors believe that based on the determined tracks and associated motion models, a species classification algorithm could be developed. In future research, the authors plan to develop an algorithm allowing to automatically determine the hot spots of the birds’ activity at airport runways, which would not be possible without this study.

## Figures and Tables

**Figure 1 sensors-22-03660-f001:**
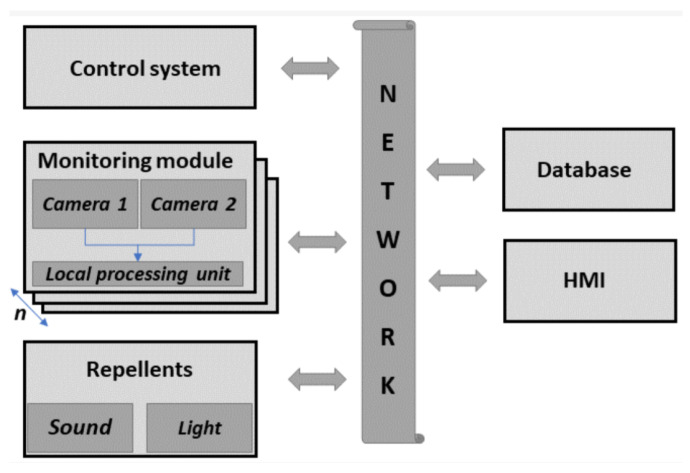
Data collection system block diagram [3].

**Figure 2 sensors-22-03660-f002:**
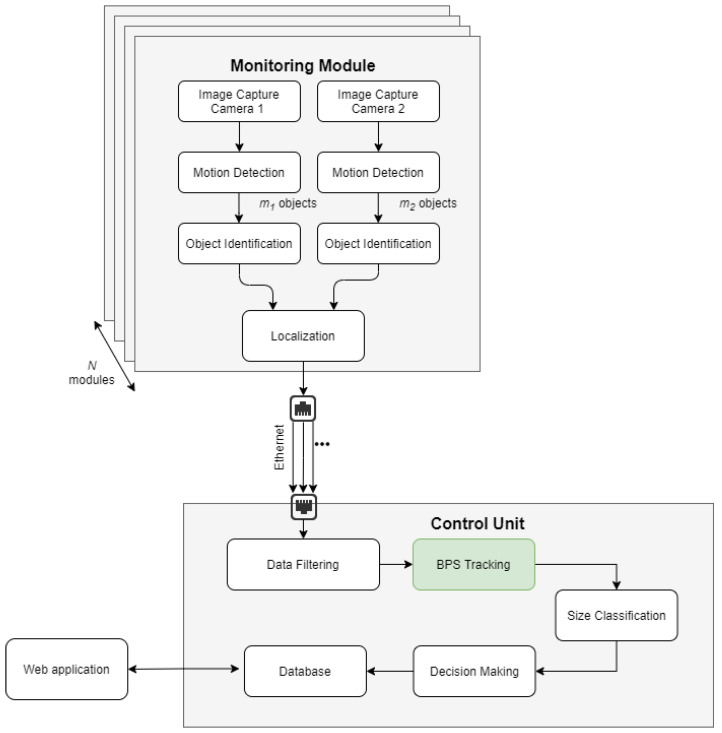
Processing architecture, based [3].

**Figure 3 sensors-22-03660-f003:**
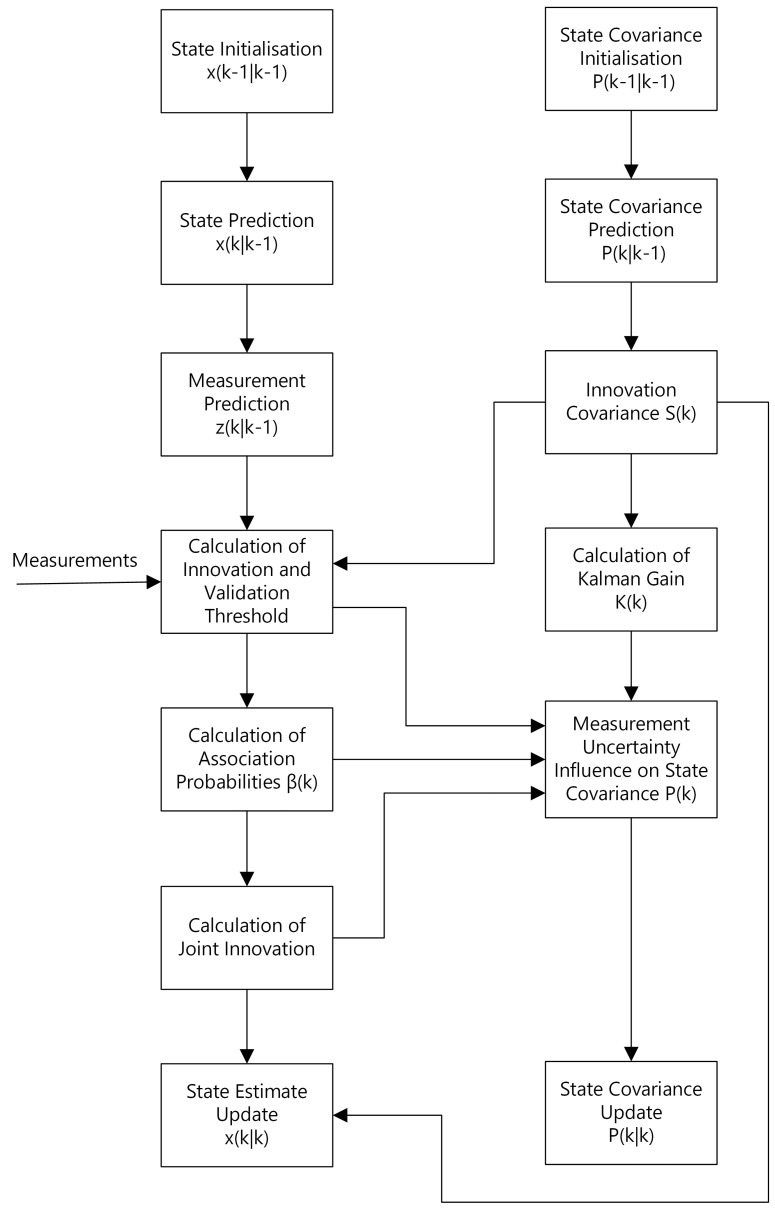
Probabilistic Data Association (PDA) algorithm block diagram based on [10].

**Figure 4 sensors-22-03660-f004:**
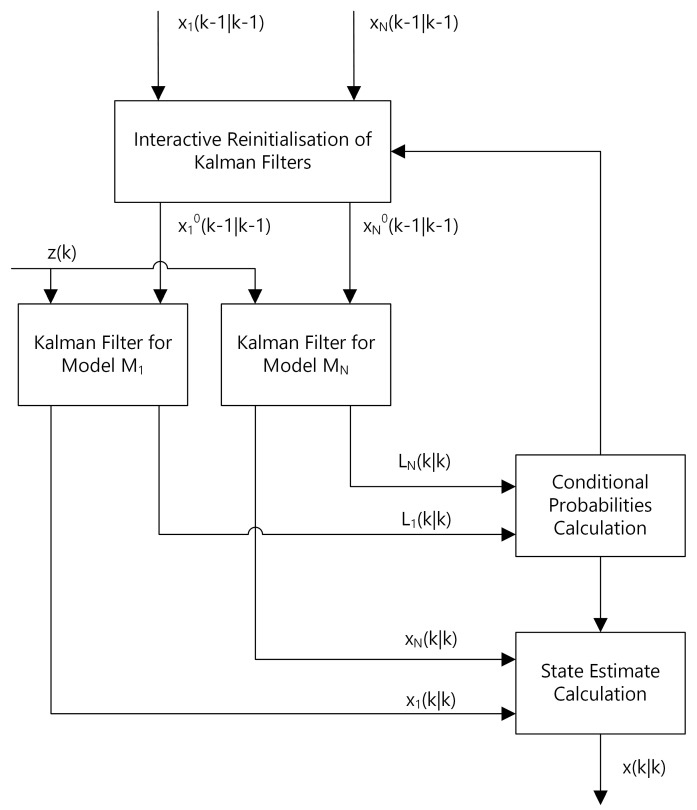
Interacting Multiple Model (IMM) algorithm block diagram [33,34].

**Figure 5 sensors-22-03660-f005:**
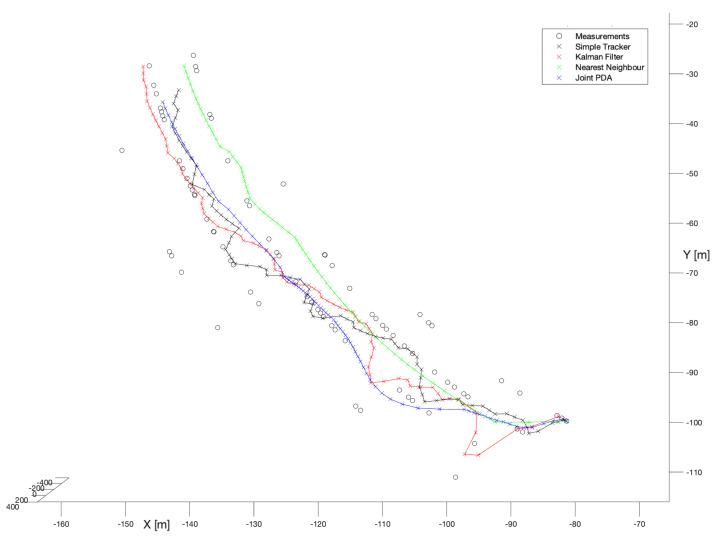
Performance of all trackers.

**Figure 6 sensors-22-03660-f006:**
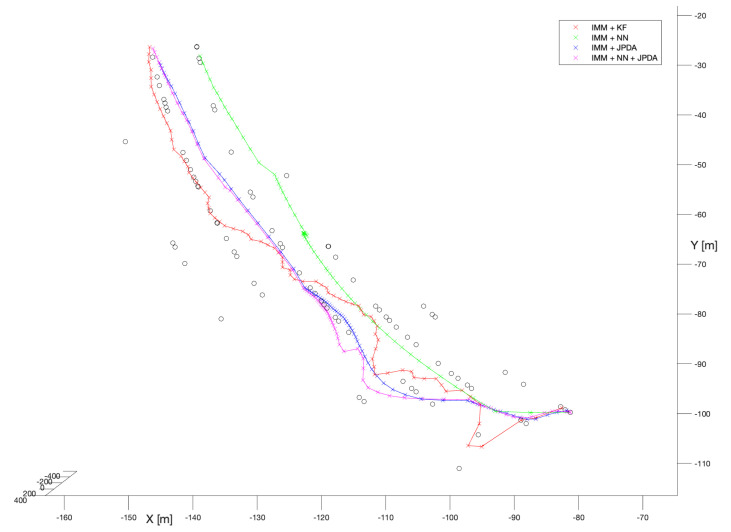
Interacting Multiple Model filters combined with: KF, NN, JPDA, and JPDA + NN, unknown process noise covariance.

**Figure 7 sensors-22-03660-f007:**
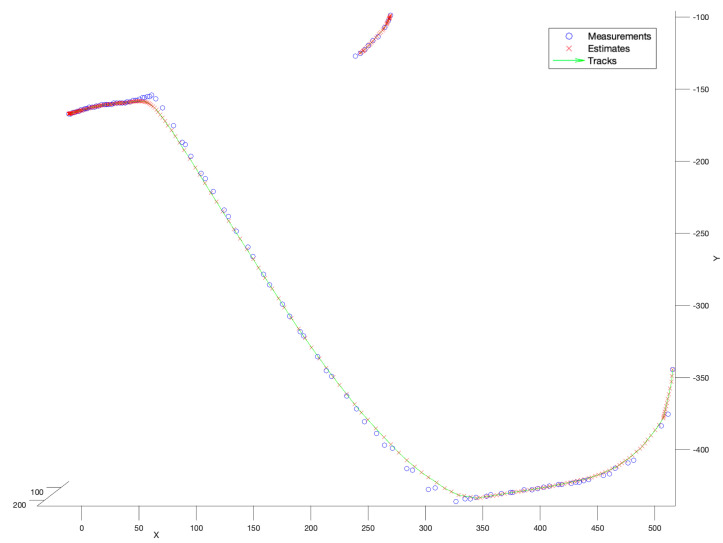
Performance of no-model tracker. Multiple bird flight scenario.

**Figure 8 sensors-22-03660-f008:**
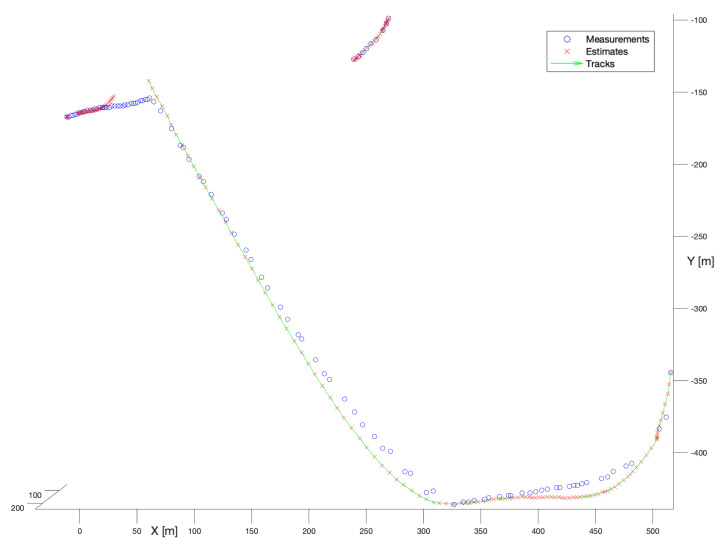
Kalman Filter, known process noise covariance. Multiple bird flight scenario.

**Figure 9 sensors-22-03660-f009:**
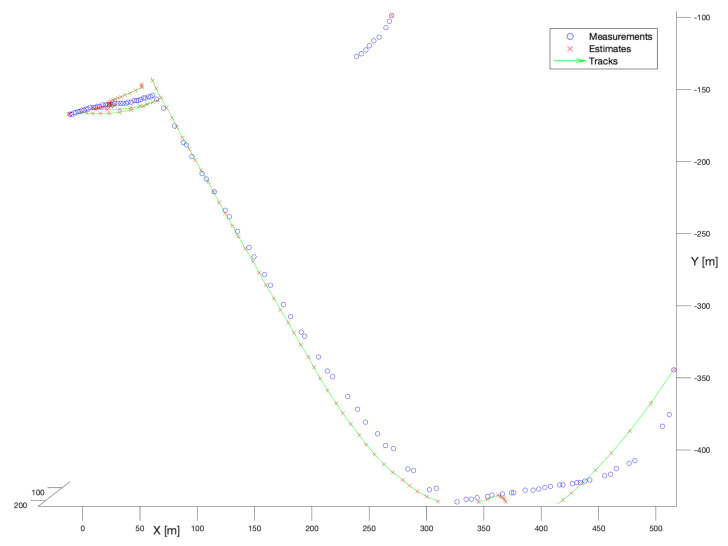
Kalman Filter + Nearest-Neighbour. Multiple bird flight scenario.

**Figure 10 sensors-22-03660-f010:**
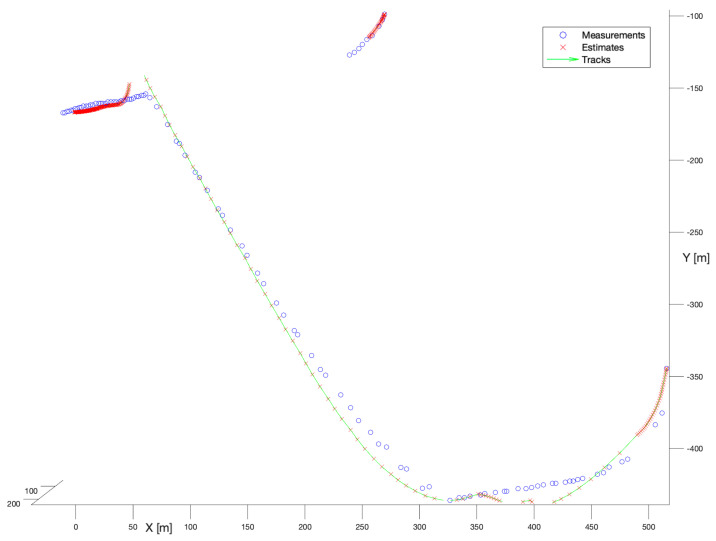
Kalman Filter + Joint Probabilistic Data Association. Multiple bird flight scenario.

**Figure 11 sensors-22-03660-f011:**
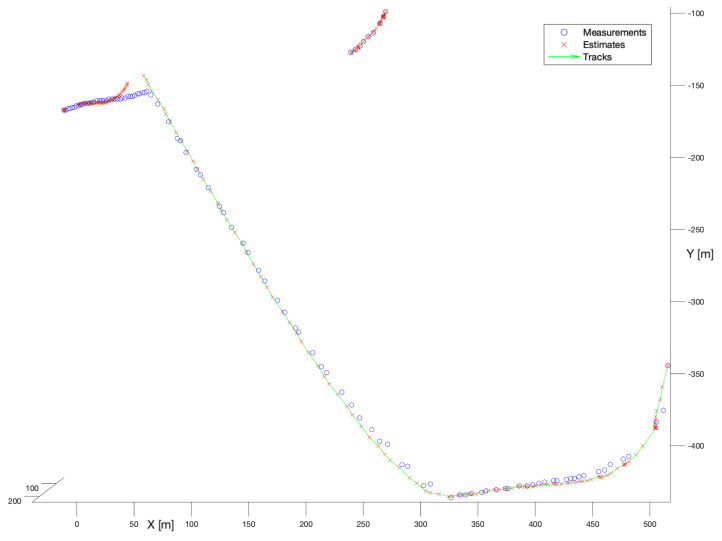
Multiple tracks Interacting Multiple Model. Multiple bird flight scenario.

**Figure 12 sensors-22-03660-f012:**
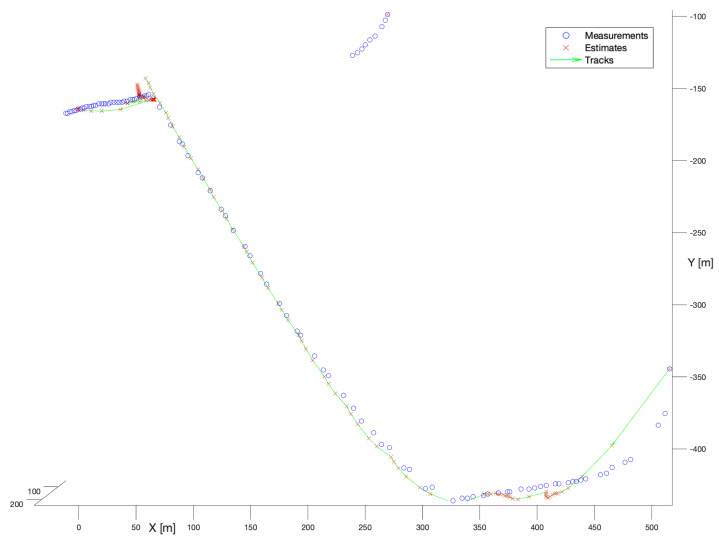
Interacting Multiple Model + Nearest-Neighbour. Multiple bird flight scenario.

**Figure 13 sensors-22-03660-f013:**
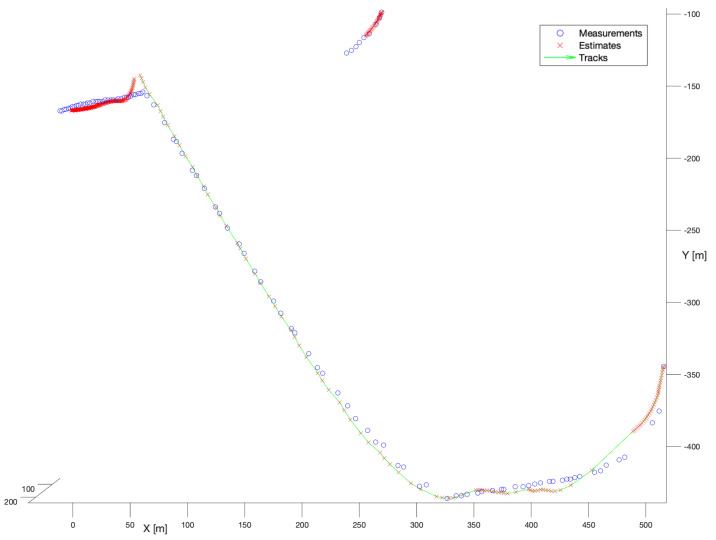
Interacting Multiple Model + Joint Probabilistic Data Association. Multiple bird flight scenario.

**Figure 14 sensors-22-03660-f014:**
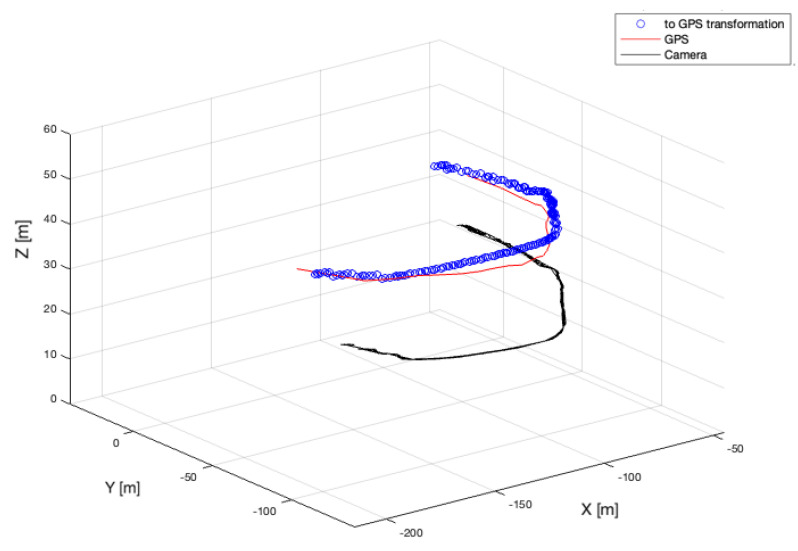
Transformation of camera coordinate system into GPS coordinate system.

**Figure 15 sensors-22-03660-f015:**
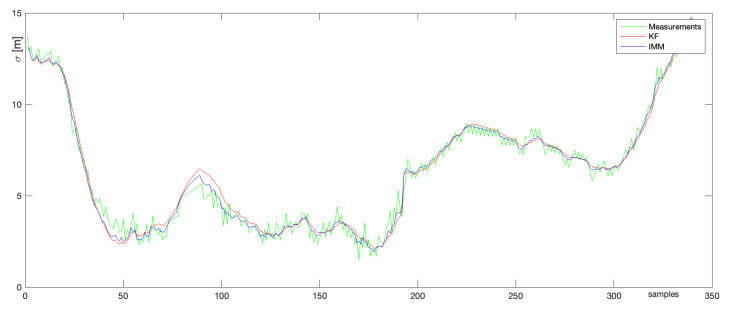
Comparison of estimation errors.

**Table 1 sensors-22-03660-t001:** Summary of the examined filters’ features.

Filter Name	Kinematic Target Model			False Alarm Dealing		
	None	Fixed	Adaptive	None	Simple	Probabilistic
Simple Filter	✓	-	-	✓	-	-
KF	-	✓	-	✓	-	-
NN	-	✓	-	-	✓	-
JPDA	-	✓	-	-	-	✓
IMM	-	-	✓	✓	-	-
IMM + NN	-	-	✓	-	✓	-
IMM + JPDA	-	-	✓	-	-	✓
IMM + NN + JPDA	-	-	✓	-	-	✓

**Table 2 sensors-22-03660-t002:** Summary of the conclusions drawn from the experiments.

Scenario	Process Noise Covariance	Track Continuity	Track Smoothing
Single bird	Known	All satisfactory	JPDA
	Unknown	All satisfactory	JPDA
Multiple birds	Known	No-model tracker	KF
	Unknown	IMM-KF	IMM-KF
Drone	Known	KF	KF
	Unknown	IMM-KF	IMM-KF

## Data Availability

Presented data accessible for authorised staff accordingly to local regulation.

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
