# Peer review of "Application of Radar Solutions for the Purpose of Bird Tracking Systems Based on Video Observation"

_sensors, 2022, doi:10.3390/s22103660_

Round 1
Reviewer 1 Report
The aim of this paper is to review tracking algorithms widely used in Radar Science, and assess possibilities of application of these algorithms for the purpose of tracking birds with video cameras. I have the following comments:
1, The English expression should be improved greatly.
2, The units of the x-axis and y-axis of Fig. 5-15 are not given.
3, The bird target is tracked in a 3D coordinate system. Why the tracked trajectory given in Fig. 5-Fig. 13 are shown in a 2D coordinate system?
4, There are many problems with the format of the article, especially on pages 20 and 21.
5, In order to express clearly, it is suggested to list and summarize the conclusions drawn from the simulations in a table.
6, The radar tracking algorithm discussed in this paper is mainly aimed at radar targets. However, the difference between bird target and other radar targets is not shown clearly, and there is a lack of theoretical analysis on whether these radar tracking algorithms are suitable for bird targets.
7, Why not improve the existing radar tracking algorithm for bird targets to propose a tracking algorithm suitable for bird targets?
Author Response
Respectable Reviewer,
Thank you very much for your in-depth analysis and your valuable feedback that allows significantly improve the quality of our article. All your remarks have been addressed in the present version of the paper and all changes are marked in yellow.
In reference to your detailed suggestions:
1. The English expression should be improved greatly.
Response 1. We have read the article carefully and improved the English usage. If in your opinion still an improvement is required, we could make the commitment to enlist the help of the MDPI’s native speaker.
2. The units of the x-axis and y-axis of Fig. 5-15 are not given.
Response 2. The units have been added.
3. The bird target is tracked in a 3D coordinate system. Why the tracked trajectory given in Fig. 5-Fig. 13 are shown in a 2D coordinate system?
Response 3. To ensure the visibility of the data and to allow the reader to appreciate the results we decided to present the tracks in the 2D coordinates.Please note that there is not much change in the state vector corresponding object height. Therefore, the presentation of the bird motion in 3D coordinate system would obscure the important manoeuvres that are perfectly visible in 2D.
4. There are many problems with the format of the article, especially on pages 20 and 21.
Response 4. The style and formatting have been improved.
5. In order to express clearly, it is suggested to list and summarize the conclusions drawn from the simulations in a table.
Response 5. We do agree that presenting the conclusion in the table improves the visibility of the paper.
6. The radar tracking algorithm discussed in this paper is mainly aimed at radar targets. However, the difference between bird target and other radar targets is not shown clearly, and there is a lack of theoretical analysis on whether these radar tracking algorithms are suitable for bird targets.
Response 6. and 7. According to the best of our knowledge, there is a lack of suitable tracking algorithms for large-base stereovision systems designed to surveil the vast monitoring area (in a radius of 500m). Therefore, despite of some differences, we decided to apply and tune the radar-based algorithms. Indeed, a typical radar target differs from a bird one in many aspects, from which the size and kinematics are the most important. There are also many differences at the sensors level such as the measurement uncertainty and sampling rate.
However, we claim our hypothesis that the processing of the bird data obtained from the stereovision system is is mostly the same as the processing of radar target data. By adjustment of standard deviation of the process noise, standard deviations of the observation noise, state model, number of local filters used in IMM, transition probabilities values in IMM we could develop the robust bird tracking algorithm. We believe that provided results support our hypothesis.
7. Why not improve the existing radar tracking algorithm for bird targets to propose a tracking algorithm suitable for bird targets?
Response 7. Please find the Response 6. It covers 6. and 7.
Reviewer 2 Report
The manuscript “Application of Radar Solutions for the Purpose of Bird Tracking System based on Video Observation”, by Ksawery Krenc ,Dawid Gradolewski, Damian Dziak, and Adam Kawalec, offer an approach of radar solutions algorithms on video birds tracking. The manuscript is not well arranged, meaning that, authors add affiliations seems to be missing (see my comments below), after conclusion sections, the authors did not complete the sections with their own information. Also, the introduction is poor, and needs to be completed.
The manuscript can be considered for publication, after Major revisions.
- Please review the affiliations; it seems that affiliation 3 is missing;
- At the correspondence e-mail is written F.L., but it is not in the authors list;
- Also, it seems some authors contributed equally to this work: which one?
- Introduction section is poor, and contains only 1 reference; please enhance the quality of the introduction section.
- Please define more accurately your main idea of the manuscript. So far, it seems that you tried to detect birds, using a video camera, and you explained the tracking algorithms involved in this process. Also, please add some words about the accuracy of this experimental set-up, meaning, what happens if the birds don’t show up in your geo-area, and instead of birds, you will have some airplanes which at long distance they look like birds? do you have specific sensors for long range detection?
- Do you have the possibility to select specific birds? Or you take all your birds? Also, if you have in the same frame a bird along with an airplane (it happens that in some altitudes, the planes will have quit the same dimensions with a usually bird), how you manage this issue?
- Please insert the author contributions and the funding.
Author Response
Respectable Reviewer,
Thank you very much for your in-depth analysis and your valuable feedback allowing to significantly improve the quality of the paper. All your remarks have been addressed in the present version of the paper and all changes are marked in yellow.
In reference to your detailed suggestions:
1. Please review the affiliations; it seems that affiliation 3 is missing
Response 1. We have reviewed and corrected all the affiliations.
2. At the correspondence e-mail is written F.L., but it is not in the authors list;
Response 2. We have reviewed and corrected all the e-mail addresses.
3. Also, it seems some authors contributed equally to this work: which one?
Response 3. We have verified and corrected the contributions.
4. Introduction section is poor, and contains only 1 reference; please enhance the quality of the introduction section.
Response 4. We have improved the introduction section and provided more references.
5. Please define more accurately your main idea of the manuscript. So far, it seems that you tried to detect birds, using a video camera, and you explained the tracking algorithms involved in this process. Also, please add some words about the accuracy of this experimental set-up, meaning, what happens if the birds don’t show up in your geo-area, and instead of birds, you will have some airplanes which at long distance they look like birds? do you have specific sensors for long range detection?
Response 5. The description of the main idea is now extended. The motivation is highlighted in chapter 1. “Introduction” and the main contribution, hypothesis, and objectives of the paper are addressed in chapter 3. The need for the paper came from the real application. State-of-the-art bird monitoring algorithms allow monitoring of the airports' runway and are widely used in Wildlife Hazard Management. The next natural step of such a system is to track and classify the targets separately. This would allow performing long-term observation and too early in advance determine the potential threat. According to the best of our knowledge, there is a lack of suitable tracking algorithms for large-base stereovision systems designed to surveil the vast monitoring area (in a radius of 500m). Therefore, in the presented paper, we address this need by adopting radar tracking algorithms. The accuracy of the system is addressed in our previous work, please see
[1] Gradolewski, D.; Dziak, D.; Kaniecki, D.; Jaworski, A.; Skakuj, M.; Kulesza, W.J. A Runway Safety System Based on Vertically Oriented Stereovision. Sensors 2021, 21, 1464. https://doi.org/10.3390/s21041464
[2] Gradolewski, D.; Dziak, D.; Martynow, M.; Kaniecki, D.; Szurlej-Kielanska, A.; Jaworski, A.; Kulesza, W.J. Comprehensive Bird Preservation at Wind Farms. Sensors 2021, 21, 267. https://doi.org/10.3390/s21010267
6. Do you have the possibility to select specific birds? Or you take all your birds? Also, if you have in the same frame a bird along with an airplane (it happens that in some altitudes, the planes will have quit the same dimensions with a usually bird), how you manage this issue?
Response 6. In the current version all the bird targets obtained form the video camera are being tracked. If by any chance an airplane falls into a validation gate it can only be eliminated based on its kinematics. However, in the current software version that feature does not work flawlessly due to the imprecise bird models.
7. Please insert the author contributions and the funding.
Response 7. The authors contributions as well as funding has been inserted.
Round 2
Reviewer 2 Report
The authors have responded to all reviewer's comments and suggestions. The manuscript can be accepted for publication.